# Molecular Characterization of a Novel *Endornavirus* Conferring Hypovirulence in Rice Sheath Blight Fungus *Rhizoctonia solani* AG-1 IA Strain GD-2

**DOI:** 10.3390/v11020178

**Published:** 2019-02-20

**Authors:** Li Zheng, Canwei Shu, Meiling Zhang, Mei Yang, Erxun Zhou

**Affiliations:** 1Integrative Microbiology Research Centre, Guangdong Province Key Laboratory of Microbial Signals and Disease Control, College of Agriculture, South China Agricultural University, Guangzhou, Guangdong 510642, China; zhenglihappy0617@126.com (L.Z.); shucanwei@scau.edu.cn (C.S.); meilingzhangsy@163.com (M.Z.); mayyang@scau.edu.cn (M.Y.); 2College of Plant Protection, Hainan University/Key Laboratory of Green Prevention and Control of Tropical Plant Diseases and Pests, Ministry of Education, Haikou 570228, China

**Keywords:** mycovirus, *Endornavirus*, hypovirulence, *Rhizoctonia solani*, rice sheath blight

## Abstract

The complete sequence and genome organization of a novel *Endornavirus* from the hypovirulent strain GD-2 of *Rhizoctonia solani* AG-1 IA, the causal agent of rice sheath blight, were identified using a deep sequencing approach and it was tentatively named as *Rhizoctonia solani* endornavirus 1 (RsEV1). It was composed of only one segment that was 19,936 bp in length and was found to be the longest endornavirus genome that has been reported so far. The RsEV1 genome contained two open reading frames (ORFs): ORF1 and ORF2. ORF1 contained a glycosyltransferase 1 domain and a conserved RNA-dependent RNA polymerase domain, whereas ORF2 encoded a conserved hypothetical protein. Phylogenetic analysis revealed that RsEV1 was phylogenetically a new endogenous RNA virus. A horizontal transmission experiment indicated that RsEV1 could be transmitted from the host fungal strain GD-2 to a virulent strain GD-118P and resulted in hypovirulence in the derivative isogenic strain GD-118P-V1. Metabolomic analysis showed that 32 metabolites were differentially expressed between GD-118P and its isogenic hypovirulent strain GD-118P-V1. The differential metabolites were mainly classified as organic acids, amino acids, carbohydrates, and the intermediate products of energy metabolism. Pathway annotation revealed that these 32 metabolites were mainly involved in pentose and glucuronate interconversions and glyoxylate, dicarboxylate, starch, and sucrose metabolism, and so on. Taken together, our results showed that RsEV1 is a novel *Endornavirus*, and the infection of virulent strain GD-118P by RsEV1 caused metabolic disorders and resulted in hypovirulence. The results of this study lay a foundation for the biocontrol of rice sheath blight caused by *R. solani* AG1-IA.

## 1. Introduction

Mycoviruses (fungal viruses) are widespread in almost all major groups of Kingdom Fungi, and an increasing number of novel mycoviruses have been reported [1,2,3,4]. The presence of most mycoviruses in fungi does not cause any visible abnormal symptoms (cryptic infections) for their fungal hosts. However, some mycoviruses are known to cause phenotypic alterations, including hypovirulence and debilitation of their fungal hosts [3,5], which may be exploited as biological control agents against fungal diseases, e.g., the +ssRNA mycovirus Cryphonectria hypovirus 1 (CHV1) against *Cryphonectria parasitica* [5] and the ssDNA mycovirus Sclerotinia sclerotiorum hypovirulence-associated DNA virus 1 (SsHADV-1) against *Sclerotinia sclerotiorum* [6].

Mycoviruses in the family *Endornaviridae* commonly contain non-encapsidated RNA genomes that range from 9.8 kb to 17.6 kb in size, and there is always only one ORF encoding a putative polyprotein containing a conserved RdRp domain, as well as RNA helicase 1 (Hel-1), methyltransferase (MTR), and glycosyltransferase (GT) domains [7]. Nevertheless, only the RdRp domain is omnipresent in all endornavirus members. Endornaviruses can infect plants, fungi, and oomycetes, which can persist in their hosts over multiple generations [7]. In plant hosts, endornaviruses rely on vertical transmission through infected pollen and ova and do not move from cell to cell because of the lack of movement proteins [8]. In fungal hosts, they transmit vertically via spores and horizontally via hyphal anastomosis [9,10,11]. Most endornaviruses do not cause any apparent symptoms in their hosts. However, Vicia faba endornavirus (VfEV-447) is associated with cytoplasmic male sterility in *Vicia faba* [12], and *Helicobasidium mompa* endornavirus 1-670 (HmEV1-670) decreased virulence of the violet root rot fungus *Helicobasidium mompa* [13].

The basidiomycetous fungus *Rhizoctonia solani* Kühn [teleomorph: *Thanatephorus cucumeris* (Frank) Donk] is a notorious soil-borne plant pathogen which has a wide host range, including field crops, vegetable crops, ornamentals, and tree plants worldwide [14]. *R. solani* consists of at least 14 genetically isolated anastomosis groups (AGs) [15]. The presence of mycovirus in *R. solani* was first reported by Castanho [16]. Indirect evidence indicated that M1 (6.4 kb) and M2 (3.6 kb) are associated with enhanced or diminished virulence in *R. solani* AG-3 [17]. Further study showed that M2 dsRNA influenced the pathogenicity of the fungal host, representing a phenomenon referred to as hypovirulence [18]. More and more studies have indicated that mycoviruses can be commonly isolated from the 14 independent AGs of *R. solani* [19,20]. At present, only five viruses have been reported in *R. solani* AG-1 IA, which caused seriously rice sheath blight in rice growing areas [14,21,22,23,24]. Previous studies have mainly focused on the isolation, sequencing, and genome structure analysis of mycoviruses. Until recently, little was known about the interaction mechanisms between mycoviruses and their fungal hosts.

Here, the discovery of a novel *Endornavirus* (*Rhizoctonia solani* endornavirus 1, RsEV1) in a hypovirulent strain GD-2 of *R. solani* AG-1 IA, the causal agent of rice sheath blight, is described in detail. Firstly, the genome organization and phylogeny of RsEV1 are analysed. Secondly, the RsEV1 is transmitted to a mycovirus-free virulent strain GD-118P via the dual culture technique, and the metabolomic analysis of the virulent strain GD-118P and its derivative isogenic strain GD-118P-V1 acquired RsEV1 from the hypovirulent strain GD-2 is performed so as to understand the effects of RsEV1 on its fungal host. Finally, the effects of viral infection on the phenotype and virulence level of the RsEV1 recipient strain GD-118P-V1 of *R. solani* AG-1 IA are evaluated.

## 2. Materials and Methods

### 2.1. Fungal Strains and Cultural Conditions

Strain GD-2 of *R. solani* AG-1 IA, which contains viral dsRNA, was isolated from rice sheath with blight symptoms in Lechang county, Guangdong province, China, and had a slower growth rate and reduced pathogenicity when compared with virulent strains. Strain GD-118P, a mycovirus-free virulent hygromycin-resistant strain maintained in our laboratory, was used as a mycovirus recipient in the horizontal transmission experiment. All strains used in this study were maintained on potato dextrose agar (PDA) medium at 28 °C and stored on PDA slants at 4 °C.

### 2.2. Extraction of dsRNA

DsRNAs were extracted from 10 g of frozen mycelia by selective absorption to the columns of cellulose powder CF-11 (Whatman, UK) according to the method described by Morris and Dodds [25], with minor modifications. For the preparation of mycelium from strain GD-2, a mycelial agar plug 5 mm in diameter cut from a colony margin of a two-day-old culture was inoculated onto each PDA plate covered with a layer of cellophane membrane and cultured at 28 °C for 5 d. The harvested mycelia were stored at −80 °C until required.

After extraction, dsRNAs were further treated with DNase I and S1 nuclease (TaKaRa Bio Inc, China), which can digest the contaminated genomic DNA and ssRNA, respectively, and the quality of purified dsRNAs was then evaluated via 1.0% (*w/v*) agarose gel electrophoresis.

### 2.3. cDNA Library Construction, Illumina Sequencing, and Sequence Analysis

The dsRNA sample was used for cDNA library construction using the NEBNext^®^Ultra^TM^ RNA Library Prep Kit (Illumina, USA), following the manufacturer’s instructions. The cDNA is end-repaired and adenylated prior to adaptor ligation, library construction, and amplification when using this method. Then, the sequence-ready library was subjected to 8 million of 150 nucleotide (nt) paired-end reads using Illumina Hiseq 4000 technology. The cDNA library construction and deep sequencing analysis were carried out by Shanghai Hanyu Bio-Tech Co., Ltd. In total, 56 contigs were obtained and the N50 length was 1466 nt. The minimum contig length was 500 bp and the minimum coverage was 18. The GC content of the full-length sequence was about 42.37%. The ends and the other parts of the sequences were all confirmed by Sanger sequencing. To obtain the terminal sequences of the virus, rapid amplification of cDNA ends (RACEs) was conducted [21].

Open reading frames (ORFs) were determined using the National Center for Biotechnology Information (NCBI) ORF Finder program (http://www.ncbi.nlm.gov/gorf/gorf.html). The program of BLASTp in the NCBI database was used to search for the conserved domain and similar sequences of the deduced amino acid sequences. Motif searches were performed in three databases, i.e., the PROSITE database (http://www.expasy.ch/), the Pfam database (http://pfam.sanger.ac.uk/), and the CDD database (http://www.ncbi.nlm,nih.gov/Structure/cdd/wrpsb.cgi). Phylogenetic trees were constructed based on the deduced amino acid sequences of the RdRp regions using the maximum-likelihood (ML) method of the Molecular Evolutionary Genetics Analysis (MEGA) software version 6.0 with 1000 bootstrap replicates, as described previously, with minor modifications [26]. Multiple alignments of the sequences of RdRp were performed with the Clustal-X program [27].

### 2.4. Transmission of Mycovirus

The dual culture technique [28] was used to test the transmissibility of the hypovirulent traits and the dsRNA element from a hypovirulent strain to a virulent strain of *R. solani*. The mycovirus-containing hypovirulent strain GD-2 was used as the viral donor, and the mycovirus-free virulent hygromycin-resistant strain GD-118P was used as the viral recipient. These two stains, GD-2 and GD-118P, were cultured in the same Petri dish several centimeters apart and were allowed to undergo hyphal fusion or anastomosis. After seven days, the mycelium from the zone of anastomosis was sub-cultured twice on hygromycin-containing PDA plates.

### 2.5. Metabolomics Analysis

The virulent strain GD-118P (mock) and the derivative isogenic strain GD-118P-V1 (treatment), which received endornavirus RsEV1 from the hypovirulent strain GD-2, were used for metabolomic analysis. After five days of cultivation on PDA plates covered with an autoclaved cellophane membrane at 28 °C, fungal mycelia from eight independent replicates were collected using a sterilized scalpel.

In total, 100 mg of each frozen fungal sample was homogenized with 500 μL pre-cooled pure methanol and vortex for 30 s. The mixes were soaked in liquid nitrogen for 15 min and then thawed on ice for 15 min, followed by a centrifugation of 13,000 rpm for 5 min at 4 °C. The supernatant was transferred to a new tube A. The debris of fungal tissues was suspended with 250 μL Milli-Q water and homogenized using an equal quantity of Sigma glass beads. After centrifugation, the supernatant was added into tube A and then blow-dried by nitrogen. In total, 60 μL of 15 mg/mL methoxamine hydrochloride was added into the precipitate, mixed well with a vortex mixer for 1 min, and then incubated overnight at room temperature. After that, 60 μL of methyl-trimethyl-silyl-trifluoroacetamide (with 1% chlorotrimethylsilane) was added, vortexed for 1 min, and incubated for 1 h at room temperature. The mixture was centrifuged at 13,000 rpm for 5 min and 100 μL supernatant was used for GS-MS analysis. The GC-MS was carried out on an Agilent 7890A/5975C Gas Chromatograph Mass Spectrometer with an HP-5MS Crossbond 5% phenyl methyl silox capillary column (0.25 mm × 30 mm × 0.25 μm, Agilent J&W scientific, Folsom, CA, USA).

The raw data from the Agilent 7890A/5975C Gas Chromatograph Mass Spectrometer were preprocessed using XCMS software (www.bioconductor.org) [29]. Unsupervised principal component analysis (PCA) and supervised partial least squares discriminant analysis (PLS-DA) were performed on the clean data by SIMCA-P version 11.0 (Umetrics, Sweden). Potential targets were identified in the NIST 2008mass spectra database and Wiley 9 Metabolomics database [30]. Pathway annotation was performed by the MetaboAnalyst 4.0 platform [31].

### 2.6. Statistical Analysis

Analysis of variance in SAS V8.0 (SAS Institute, NC, USA) was performed to analyse the data on sclerotium dry weight, mycelial growth rate, and leaf lesion area caused by strains of *R. solani*. Six replicative plates were made for each strain and each experiment was repeated three times.

## 3. Results

### 3.1. Nucleotide Sequence and Amino Acid Sequence of RsEV1

The complete genome of RsEV1 was 19,936 bp in length, with a GC content of 42.37% (Figure 1). The full-length cDNA sequence of RsEV1 was deposited in the GenBank database under the accession no. MF098782. The 5’- and 3’-untranslated regions of the dsRNA were 9 and 50 nt in length (Figure 1). Analysis of RsEV1 organisation indicated that it contains two putative ORFs (ORF1 and ORF2), unlike the common endornaviruses containing only one ORF. The presence of ORF2 was confirmed using PCR and Sanger sequencing. The sequences between two termination codons were amplified with specific primers based on the 3’ ends of ORF1 and ORF2 and sequenced with the Sanger method. The results showed that sequences obtained from Sanger and Illumina sequencing had 100% identity. It is noteworthy that there is an intergenic spacer of 278 nt between ORF1 and ORF2. The ORF1, starting at nt 10 and ending at nt 16,624, potentially encodes a 5538-amino-acid (aa) protein with a predicted molecular mass of 624.91 kDa (Figure 1). Sequence analysis using a conserved domain database (CDD) indicated the presence of two protein domains, GT1 and RdRp, in ORF1. Furthermore, multiple protein alignment showed that the predicted RdRp domain includes four conserved motifs (A to D) similar to those of *Helicobasidium mompa* alphaendornavirus 1 (HmEV1) and *Ceratobasidium* endornavirus C (CbEVC) (Figure 2). The ORF2, starting at nt 16,901 and ending at nt 19,886, potentially encodes a 995 aa protein with a predicted molecular mass of 113.23 kDa (Figure 1). A BLASTP search of the deduced aa sequence of RsEV1 confirmed that ORF2 encodes a hypothetical protein that has 29% identity to CbEVC.

### 3.2. Phylogenetic Analyses

The phylogenetic tree constructed with RdRp aa sequences indicated that RsEV1 was closest to other fungus-derived endornaviruses, such as HmEV1, CbEVC, and *Rhizoctonia cerealis* alphaendornavirus 1(RcEV1) (Figure 3). A previous study showed that the endornaviruses were divided into clade I (*Alphaendornavirus*) and clade II (*Betaendornavirus*) based on the length of the genome, phylogeny of the RdRp, and host type [11]. Phylogenic analysis placed RsEV1 within Clade I. Taken together, genome organizations, amino acid sequence alignments, and phylogenetic analyses all support that RsEV1 is a new member of the genus *Endornavirus* within the family *Endornaviridae*. This is the largest endornavirus that has been reported from all known endornaviruses so far.

### 3.3. Transmission of Hypovirulence and DsRNA

Six hygromycin-resistant derivative strains obtained from the dual culture experiment were screened for the presence of viral dsRNA, but only one derivative strain GD-118P-V1 was finally found to contain the viral dsRNA. The specific 19,936 bp dsRNA segment was detected in the derivative isogenic strain GD-118P-V1, as well as in GD-2, but not in GD-118P (Figure 4). Colony morphologies of these two isogenic strains GD-118P and GD-118P-V1 grown under the same conditions were compared. The results indicated that GD-118P-V1 had thinner mycelia, smaller sclerotia, and dark pigmentation on the PDA plate when compared with GD-118P and GD-2 (Figure 5A). In addition, the effect of RsEV1 on fungal virulence was evaluated based on lesion sizes on rice leaves caused by the two isogenic strains GD-118P and GD-118P-V1 (Figure 5B). RsEV1 infection reduced the mycelial growth (Figure 5C) and resulted in the increase of sclerotial dry weight (Figure 5D) in strain GD-118P-V1. Three days after inoculation, the average lesion areas caused by GD-118P-V1 were smaller than those caused by GD-118P (Figure 5E), suggesting that RsEV1 induced hypovirulence in the virus-infected strain GD-118P-V1.

### 3.4. Overview of Metabolomic Profiling

In this study, a total of 89 known metabolites confirmed by reference standards were obtained from 16 fungal samples. They included 15 amino acids; 24 organic acids; 21 sugars, polyols, and derivatives; 6 nucleotide derivatives; 5 phenols; and 18 other metabolites. Among them, 18 metabolites involved in amino acid and carbohydrate metabolism were identified precisely according to the retention time and characteristic of fragmentation.

The result of heatmap analysis showed that three clades could be recognized using hierarchical clustering. In total, 41 metabolites were mainly present in GD-118P, 36 metabolites were the ones mainly exhibited in GD-118P-V1, and 12 metabolites were common to both the strains (Figure 6).

### 3.5. Metabolic Profile Comparison

The PCA scatter plot showed that the GD-118P are clearly separated from the GD-118P-V1 along t(1). The t(1) represents the first principal component. Samples from GD-118P were classified by positive scores, while those from GD-118P-V1 had negative scores on t(1), indicating that GD-118P and GD-118P-V1 are different from each other in metabolic patterns (Figure 7). The PCA plot showed a good clustering of samples with high goodness of fit and predictability, as indicated by the R2 and Q2 values of 0.713 and 0.413, respectively.

The result of PLS-DA analysis also showed different metabolic profiles between GD-118P and GD-118P-V1, which is consistent with the PCA plot result. The components of the PLS-DA model explained 95.0% of the variance and the cumulative Q2 variance was 74.7% for the prediction accuracy, indicating that the inter-group difference is significant (Figure 8A). PLS-DA models were further confirmed by a seven-fold cross validation, and a permutation test was applied to validate the models’ reliability rigorously (*n* = 100). The score plots showed a clear separation between GD-118P and GD-118P-V1 (Figure 8B).

### 3.6. Metabolite Changes between GD-118P and GD-118P-V1

Using the VIP values (VIP > 1) derived from the PLS-DA score and the *p* values (*p*< 0.05), 32 differentially produced metabolites in GD-118P and GD-118P-V1 were obtained. Compared with GD-118P, the level of N-acetyl glucosamine, oxalic acid, glutamine, inosine, 1,2-dihydroxy-cyclohexene, citric acid, and phenylalanine in GD-118P-V1 was increased, whereas the level of lactic acid, dodecanoic acid, succinic acid, maltose, sedoheptulose, glucose, tyrosine, serine, xylitol, myo-inositol, and thymine was significantly decreased (Table 1).

### 3.7. Pathway Annotation of the Differential Metabolites

Pathway annotation of the differential metabolites from virus-free and -containing isogenic strains was performed to further understand their biological functions. As shown in Figure 9, 32 differential metabolites were involved in the following pathways: pentose and glucuronate interconversions (*p* = 0.045195); glyoxylate and dicarboxylate metabolism (*p* = 0.060131); starch and sucrose metabolism (*p* = 0.094173); citrate cycle (TCA cycle) (*p* = 0.11289), alanine, aspartate, and glutamate metabolism (*p* = 0.11289); aminoacyl-tRNA biosynthesis (*p* = 0.12448); and phenylalanine, tyrosine, and tryptophan biosynthesis (*p* = 0.13251).

## 4. Discussion

As reported in this study, a novel *Endornavirus* RsEV1 from *R. solani* AG-1 IA strain GD-2 was identified. Sequence analysis confirmed that the RsEV1 genome contains two ORFs. To the best of our knowledge, this is the fourth endornavirus reported to contain two ORFs after CbEVB, CbEVC, and CbEVG [11]. Before this, it was considered that endornaviruses had only one ORF encoding a replicase. Sanger sequencing of the regions surrounding the 3’ end of ORF2 further confirmed that ORF2 is not an artifact of Illumina sequencing or software assembly. The origin of the ORF2 remains unclear and it may be related to a specific function in the host. Although the new virus RsEV1 breaks through the structure of the *Endornavirus* in the genome organization, we still classify the new virus RsEV1 as a member of the genus *Endornavirus* in the family *Endornaviridae* because they have many similar genome characteristics, such as containing one large polyprotein encoded by ORF1 and encoding domains of other members of the family [11].

The RsEV1 polyprotein has a GT domain, which is uncommon in fungal and plant viruses, and has only been identified in some endornaviruses and hypoviruses [7,32,33]. The lack of a GT domain in other endornarviruses indicated that the domain is not always essential in the virus replication process. Previous study indicated that the viral GT domain was acquired from hosts during evolution, possibly prior to the separation of Kingdoms, to allow endornaviruses to protect themselves against host cellular enzymes by enforcing the membrane surrounding their capsid-less dsRNAs [7,34,35].

The RdRp domain belongs to the RdRp_2 family of RNA-dependent RNA polymerases. The enzyme is involved in replication [36,37], suggesting that RsEV1, and other endornaviruses are capable of independent replication. In addition, the RdRp domain of RsEV1 has four conserved motifs (A to D) similar to the RdRps of HmEV1 and CbEVC (Figure 2), but lacks the fifth motif (E) existing in other endornarviruses, indicating that it is probably the result of coevolution between the virus and the host fungus. Based on the information about the dsRNA segment, dsRNA genetic organization, RdRp sequences, and phylogenetic analyses, we believe that RsEV1, isolated from *R. solani* AG-1 IA, is a novel mycovirus in the genus *Endornavirus* of the family *Endornaviridae*.

RsEV1 infection is very stable in the mycelia of *R. solani* AG-1 IA. An attempt to obtain a virus-free strain by hyphal tipping of virus-containing strain GD-2 was unsuccessful in our laboratory. In addition, *R. solani* does not produce any asexual spores and the sexual spores (teliospores) are still hard to induce in vitro [38]. However, the horizontal transmission of hypovirulence and the dsRNA from GD-2 to the virulent strain GD-118P was successful.

To clarify the effects of RsEV1 on its host *R. solani* AG-1 IA, metabolomic analysis with virus-free and virus-containing isogenic strains was performed. Metabolomics is the systematic study of the entire suite of metabolites in an organism to obtain the coordinated regulation of biological systems. Therefore, the metabolomic method is superior to the traditional biochemistry methods in metabolite analysis [39,40]. In this study, 89 known metabolites were identified from virulent virus-free strain GD-118P and virus-containing isogenic strain GD-118P-V1, which received RsEV1 from the hypovirulent strain GD-2. A principal component analysis (PCA) score scatter plot and partial least square-discriminate analysis (PLS-DA) score scatter plot indicated significant differences in metabolites between GD-118P and GD-118P-V1.

N-Acetyl glucosamine produced by the amino acids and their corresponding tRNA under the esterification reaction by N-Acetyl glucosamine synthase is an intermedium during the transmitting of amino acid to the carboxyl end of the peptide chain and plays an important role in protein biosynthesis [41,42]. Here, the level of serine and tyrosine was significantly decreased, whereas the level of glutamate and phenylalanine was significantly increased in GD-118P after infection by mycovirus RsEV1. These metabolites are involved in secondary metabolite biosynthesis and are closely related to N-Acetyl glucosamine biosynthesis. These results indicated that the infection of RsEV1 caused a protein metabolism disorder in GD-118P-V1.

In addition, the differential expression of citric acid and succinic acid in GD-118P-V1 hinted that TCA, the shikimate pathway, and the biosynthesis of other alkaloids were subject to different levels of impact. The level of endogenous metabolites such as tyrosine, glucose, and succinic acid significantly decreased, which also showed a low metabolism in GD-118P-V1. Additionally, the reduction of products involved in the TCA cycle may be due to the inhibition of enzyme activities in the TCA pathway. Furthermore, the reduction of glucose may be because of the energy demand increase when the GD-118P strain was infected by RsEV1.

In conclusion, we have reported for the first time that RsEV1 is a new *Endornavirus* in *R. solani*, and the infection of RsEV1 resulted in disorders of the metabolisms of amino acid, protein synthesis, and the TCA cycle of the host and caused hypovirulence. Our results not only lay a foundation for the further study of interactions between endogenous viruses and their fungal host, but also provide a potential view for the biocontrol of rice sheath blight caused by *R. solani* AG1-IA by using mycoviruses.

## Figures and Tables

**Figure 1 viruses-11-00178-f001:**
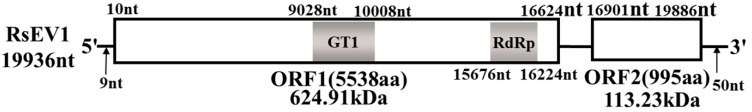
Genomic organization of a novel dsRNA endornavirus, *Rhizoctonia solani* endornavirus 1 (RsEV1) in *Rhizoctonia solani* AG-1 IA. The open reading frame (ORF) and the untranslated regions (UTRs) are indicated by a rectangle and single line, respectively. The shadowing parts indicate the glycosyltransferase 1 (GT1) and RNA-dependent RNA polymerase (RdRp) conserved domains. The corresponding nucleotides in the genome are given above the rectangle, and the amino acid numbers and the molecular masses of the proteins are shown below the rectangle.

**Figure 2 viruses-11-00178-f002:**
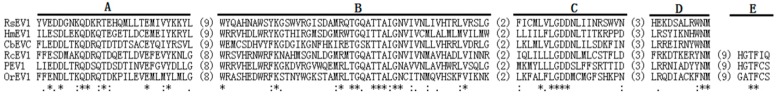
Sequence alignment of RsEV1 RdRp motifs with those of selected viruses in the genus *Endornavirus*. Horizontal lines, labeled with A to E, above the alignment indicate the five motifs, and the numbers in brackets show the amino acid positions. Asterisks, colons, and dots show the same amino acid residues, conservative, and semi conservative, respectively.

**Figure 3 viruses-11-00178-f003:**
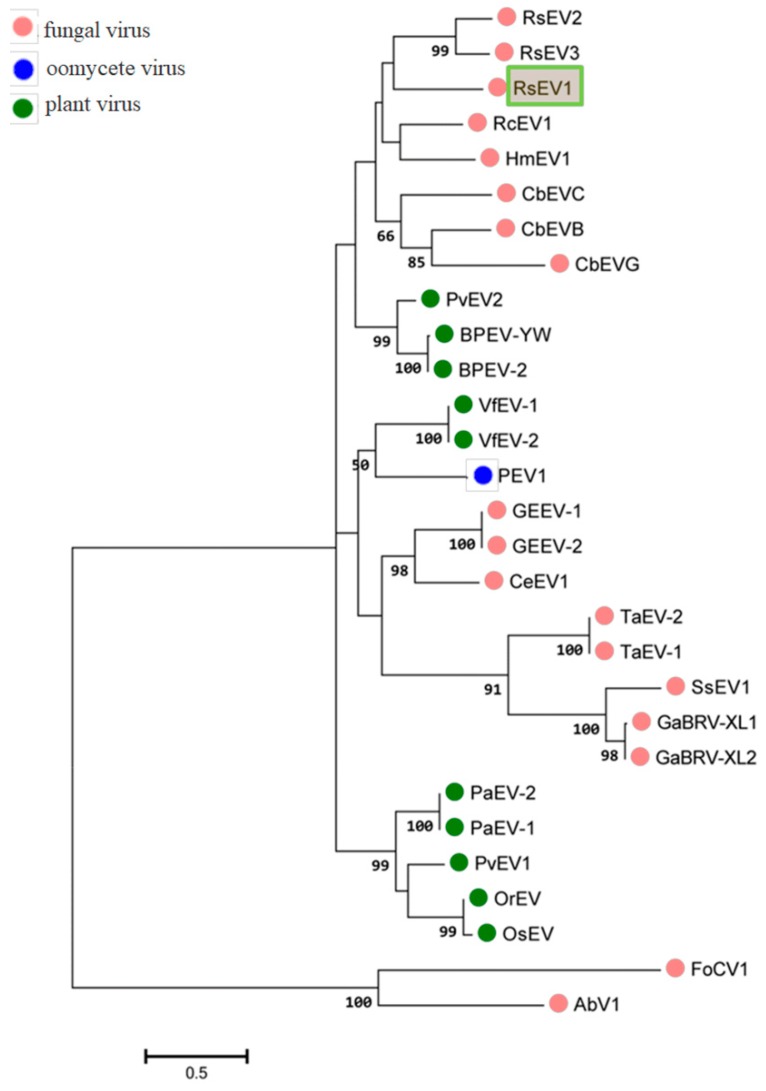
Phylogenetic analysis of RsEV1 and other endornaviruses based on the deduced amino acid sequences of RdRps using the maximum likelihood (ML) method with 1000 bootstrap replicates. The scale bar indicates a genetic distance of 0.5 amino acid substitutions per site. The green box represents the novel mycovirus RsEV1 in this study. The abbreviations of viruses and their GenBank accession numbers are as follows: RsEV2, *Rhizoctonia solani* endornavirus 2 (AMM45288.1); RsEV3, *Rhizoctonia solani* endornavirus 3 (ANR02699.1); RcEV1, *Rhizoctonia cerealis* alphaendornavirus 1 (YP_008719905.1); HmEV1, *Helicobasidium mompa* alphaendornavirus 1 (YP_003280846.1); CbEVC, *Ceratobasidium* endornavirus C (YP_009310111.1); CbEVB, *Ceratobasidium* endornavirus B (YP_009310114.1); CbEVG, *Ceratobasidium* endornavirus G (YP_009310116.1); PvEV2, Phaseolus vulgaris alphaendornavirus 2 (ATB20098.1); BPEV-YW, Bell pepper alphaendornavirus YW (AEK22062.1); BPEV-2,Bell pepper alphaendornavirus 2 (BAK52155.1); VfEV-1, Vicia faba endornavirus 1 (YP_438201.1); VfEV-2, Vicia faba endornavirus 2 (CAA04392.1); PEV1, Phytophthora alphaendornavirus 1 (YP_241110.1); GEEV-1, Grapevine endophyte alphaendornavirus 1 (YP_007003829.1); GEEV-2, Grapevine endophyte alphaendornavirus 2 (AFV91541.1); CeEV1, Chalara elegans endornavirus 1 (ADN43901.1); TaEV-2, Tuber aestivum betaendornavirus 2 (ADU64759); TaEV-1, Tuber aestivum betaendornavirus 1 (YP_004123950.1); SsEV1, Sclerotinia sclerotiorum endornavirus1(AJF94392.1); GaBRV-XL1, Gremmeniella abietina type B RNA virus XL1 (YP_529670.1); GaBRV-XL2, Gremmeniella abietina type B RNA virus XL2 (ABD73306.1); PaEV-2, Persea americana alphaendornavirus 2 (AEX28369.1); PaEV-1, Persea americana alphaendornavirus 1 (YP_005086952.1); PvEV1, Phaseolus vulgaris alphaendornavirus 1 (YP_009011062.1); OrEV, Oryza rufipogon alphaendornavirus (YP_438202.1); OsEV, Oryza sativa alphaendornavirus (YP_438200.1). AbV1 (Agaricus bisporus virus 1) and FoCV1 (Fusarium oxysporum chrysovirus 1) were used as outgroups.

**Figure 4 viruses-11-00178-f004:**
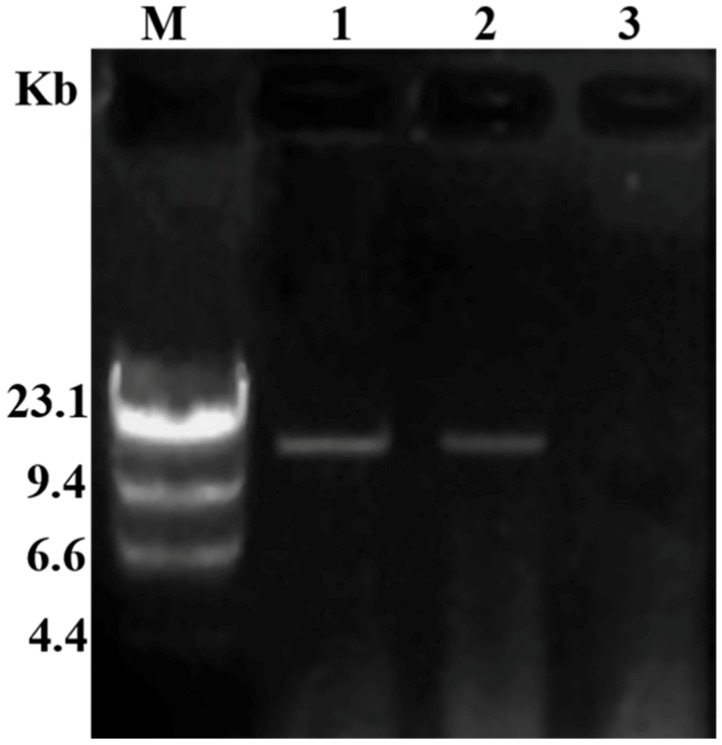
Detection of dsRNAs in viral donor and recipient strains of *Rhizoctonai solani* AG-1 IA. M: molecular markers (λ DNA digested with *Hind* III); 1: The presence of dsRNA in derivative isogenic strain GD-118P-V1; 2: The presence of dsRNA in virus-containing donor strain GD-2; 3: The absence of dsRNA in virus-free recipient strain GD-118P.

**Figure 5 viruses-11-00178-f005:**
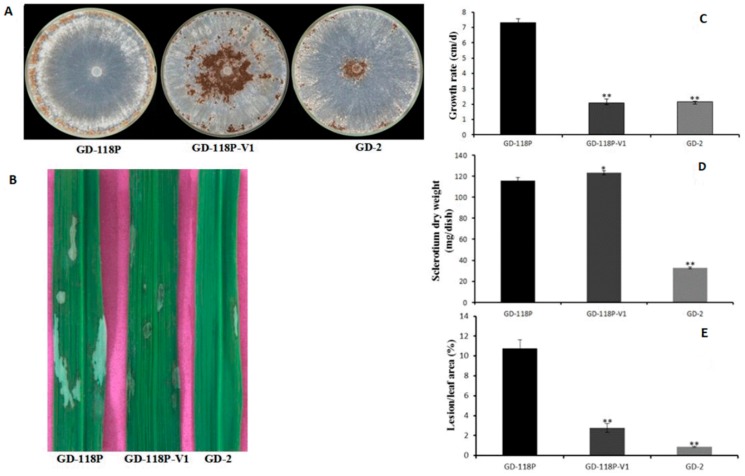
Hypovirulence-associated traits in strain GD-118P-V1 of *Rhizoctonai solani* AG-1 IA. (A) Colony morphology. Culture characteristics of strains GD-118P, GD-118P-V1, and GD-2 on PDA plates at 28 °C for seven days. (B) Pathogenicity. The symptoms on detached rice leaves caused by isogenic strains GD-118P and GD-118P-V1, and GD-2 incubated at 28 °C for 72 h. (C and D) Comparison of average mycelial growth rate and sclerotia dry weight on PDA plates of the strains GD-118P, GD-118P-V1, and GD-2. (E) Average lesion areas caused by the strains GD-118P, GD-118P-V1, and GD-2 on detached rice leaves. In (C), (D), and (E), the data were indicated as arithmetic means ± standard error, and significant differences were assessed using the Student *t* test. Single asterisk (*) indicates *p* < 0.05, and double asterisks (**) indicate *p* < 0.01.

**Figure 6 viruses-11-00178-f006:**
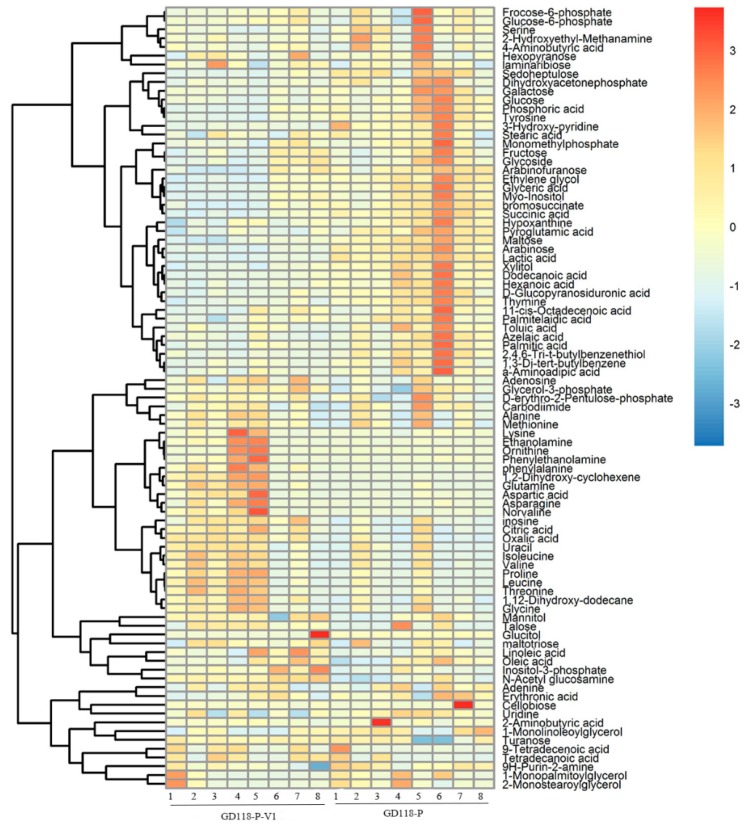
Clustering analysis of the metabolic data of isogenic fungal samples. Notes: Heat map showing the clustering of detected metabolites into three metabolite classes across 16 samples. Each row represents the level changes of a metabolite in all samples. Red colors indicate metabolite levels greater than the median value (yellow colors), and blue colors indicate metabolite levels lower than the median value (yellow colors).

**Figure 7 viruses-11-00178-f007:**
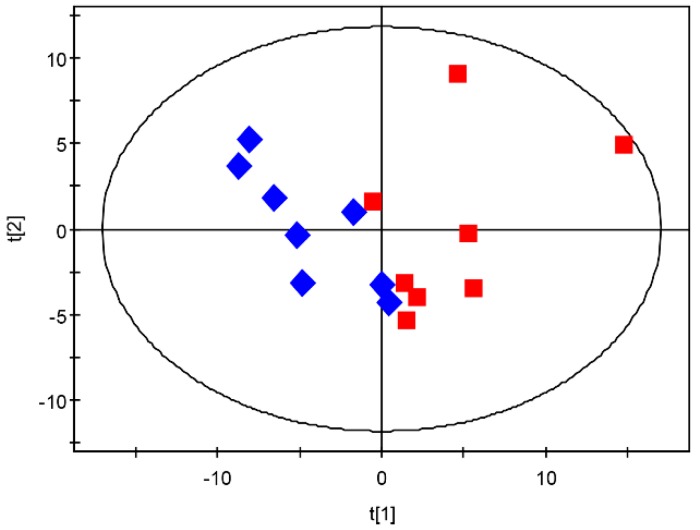
PCA scores derived from GC-MS spectra from isogenic strains GD-118P and GD-118P-V1 samples. Score plot of PCA: GD-118P (red square) and GD-118P-V1 (blue diamond).

**Figure 8 viruses-11-00178-f008:**
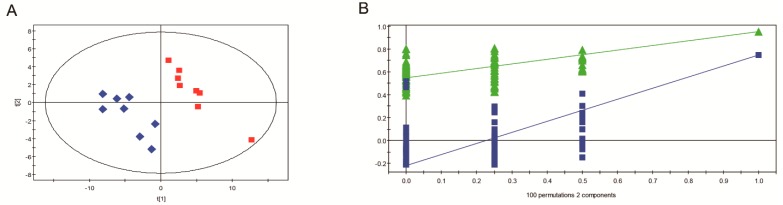
PLS-DA score plots derived from GC-MS data from isogenic strains GD-118P and GD-118P-V1 samples. A: PLS-DA score plot: GD-118P (red square) and GD-118P-V1 (blue diamond); B: Permutation test plot (100 permutations) derived from GC-MS data of GD-118P and GD-118P-V1 samples.

**Figure 9 viruses-11-00178-f009:**
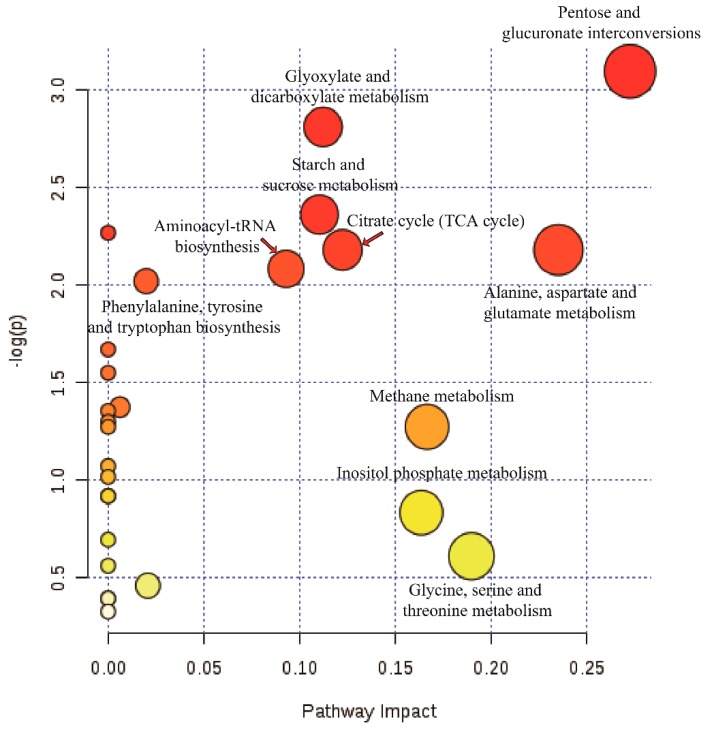
Pathway analysis of the differential metabolites from virus-free and -containing isogenic strains. Plots depicting computed metabolic pathways as a function of −log (*p*) and pathway impact for the key differential metabolites from GD-118P vs. GD-118P-V1.

**Table 1 viruses-11-00178-t001:** The differential metabolites in virus-free and -containing isogenic strains of *Rhizoctonia solani* AG-1 IA.

Item	Differential Metabolites	Retention Time (min)	VIP ^1^	*p* Value ^2^	LOG_2_(GD-118P-V1/GD-118P) ^3^
140	N-Acetyl glucosamine	23.46	2.18	0.002	0.43
19	Lactic acid	12.10	1.63	0.000	−1.04
96	Arabinose	20.74	1.43	0.000	−0.83
182	Maltose	29.57	1.40	0.000	−1.63
47	Succinic acid	15.99	1.34	0.000	−0.57
54	Thymine	16.76	1.30	0.000	−0.93
75	Pyroglutamic acid	18.75	1.30	0.000	−0.70
37	Serine	15.21	1.29	0.005	−1.28
102	Xylitol	21.22	1.27	0.001	−1.81
81	Bromosuccinate	19.05	1.26	0.002	−0.54
3	Ethylene glycol	8.87	1.25	0.003	−0.74
138	Tyrosine	23.35	1.22	0.001	−0.37
154	Sedoheptulose	25.48	1.21	0.003	−1.90
16	3-hydroxy-pyridine	11.57	1.21	0.001	−0.83
48	Glyceric acid	16.31	1.21	0.005	−0.55
93	Dodecanoic acid	20.18	1.17	0.005	−1.48
170	D-Glucopyranosiduronic acid	27.66	1.15	0.006	−0.48
80	2,4,6-Tri-t-butylbenzenethiol	19.00	1.15	0.004	−0.94
151	Myo-Inositol	24.97	1.15	0.038	−0.32
20	Hexanoic acid	12.25	1.14	0.011	−0.52
40	Phosphoric acid	15.52	1.11	0.007	−0.23
134	Glucose	23.20	1.11	0.004	−0.21
28	Oxalic acid	13.44	1.09	0.003	1.72
110	Arabinofuranose	21.67	1.08	0.035	−0.44
91	Glutamine	19.93	1.08	0.007	4.05
176	Inosine	28.92	1.06	0.007	1.57
27	1,2-Dihydroxy-cyclohexene	13.14	1.04	0.013	5.44
148	Palmitic acid	24.23	1.01	0.021	−0.55
116	Citric acid	22.22	1.01	0.006	1.58
112	Azelaic acid	21.80	1.00	0.009	−0.67
113	Hypoxanthine	21.99	1.00	0.014	−0.77
92	Phenylalanine	20.02	1.00	0.017	5.69

Notes: ^1^ indicates the variable importance in the projection; ^2^ indicates the ratio of the GD-118P-V1 and GD-118P groups; ^3^ indicates the logarithm of the ratio of the GD-118P-V1 and GD-118P groups.

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
