# Peer review of "Molecular Characterization of a Novel Endornavirus Conferring Hypovirulence in Rice Sheath Blight Fungus Rhizoctonia solani AG-1 IA Strain GD-2"

_viruses, 2019, doi:10.3390/v11020178_

Round 1

Reviewer 1 Report

No comments

Author Response

All questions have been answered directly on the PDF file. All changes in red colour have been made on the revised manuscript in word file (viruses-427824_R1). Please check the attached PDF formats.

Reviewer 2 Report

I have no major reservations but very minor ones.

Line 2. Unitalize “Endornavirus.”

Line 14. Unitalize “Endornavirus” with “E” lower-cased.

Line 433 Replace “Nobuhiro” with “Suzuki.”

Line 451. Should be “Linder-Basso.”

Author Response

Q1: Line 69. RsEV1 should be spelled out at its first appearance in the main text.

A1: Thanks! We have changed it to Rhizoctonia solani endornavirus 1.

Q2: Line 159. Should be “endornavirus.”

A2: The word ‘Endornavirus’ has been changed to ‘endornarviruses’.

Q3: Line 159-160. How did the authors confirm the sequence integrity. By sequencing clones or directly PCR fragments?

A3: This is a good question! The presence of ORF2 was confirmed using PCR andSanger sequencing methods. The sequence between two termination codons were amplified with specific primers based on the 3’ end of ORF1 and ORF2 and sequenced with Sanger method.

Q4: Lines 219-231. Did the authors confirm the symptomatology by back-introduction of RsEV1 into GD-118P?

A4: Yes! The introduction of RsEV1 into GD-118P resulted in a new isogenic strain GD-118P-V1 which contained RsEV1 and showed some hypovirulence-associated traits (such as slow growth rate, weak pathogenicity, etc.).

Q5: Line 228. Is the increase observed in GD-118P-V1 statistically significant?

A5: Yes! The differences in some biological characteristics between the two isogenic strains GD-118P and GD-118P-V1 are statistically significant.

Q6: Line 307. Why second? Not fourth?

A6: Thank you for your suggestion! We have changed ‘the second’ to ‘the fourth’.

Q7: Lines 313 and 330. There are redundant statements.

A7: Thanks! We have deleted repetitive expressions.

Q8: Line 339. It is not clear to this reviewer how “metabolomics is superior to the traditional biochemistry methods?”

A8: We have re-written the sentence so as to make it clear to reviewers/readers.

Q9: Fig. 3. The tree shows three clades rather than two (page 5). This needs to be discussed.

A9: Thank you for your suggestion! We have revised the related contents. According to the ICTV reports, previous study showed that the endornaviruses were divided into clade I (Alphaendornavirus) and clade II (Betaendornavirus) according to the length of the genome, phylogeny of the RdRp, and host type. However, more and more endornaviruses have been found so that the two genera cannot be classified well. In the present study, phylogenetic trees indicated that all endornaviruses are clustered into one branch.

Reviewer 3 Report

All my objections were answered and/or solved!

Author Response

Q1: line 41: Mycoviruses ... are .. RNA genomes?

A1: Thank you for pointing out the grammatical error. We have revised it.

Q2: line 69: RsEV1- is the first use in the plain text, should be explained

A2: Thanks for pointing out the error. We have revised it.

Q3: line 87: dsRNAs were ...purified (?) with DNase... - not precise expression

A3: Thanks for pointing out the error. The word ‘purified’ was changed to ‘treated’.

Q4: line 97: GC content (of what?) is about...

A4: Thanks for pointing out the error. ‘of the full-length sequence’ was added after ‘The GC content’.

Q5: line 131: ... precipitate with (?) a vortex? - strange

A5: Thanks for pointing out the error. We have changed ‘…with a vortex’ to ‘…mixed well with a vortex mixer’.

Q6: line 132: MSTFA, TMCS are unknown for biologist, please use full names

A6: Thank you for your suggestion. The full names have been added, i.e. MSTFA was changed to methyl-trimethyl-silyl-trifluoroacetamide, and TMCS was changed to chlorotrimethylsilane.

Q7: line 133: ...centrifuged at 13000 rpm

A7: Yes! We have changed it to the correct expression.

Q8: lines 137-142 describe specific details used in the specific device, could be omitted in the final text

A8: Thank you for your suggestion! The related contents have been deleted.

Q9: line 154. I wonder that other domains described in another endornaviruses (Helicase, MTR, Cys-rich) were not tested and discussed? see line 315: ...encoding domains...

A9: Thank you! We only detected two protein domains (GT1 and RdRp) in ORF1 of RsEV1 using conserved domain database. The absence of other domains indicated that they are not necessary for RsEV1 evolution.

Q10: line 157: UTR is used only one in whole text, should be omitted

A10: The UTR has been deleted and just used its full spelling ‘the untranslated region’.

Q11: line 159: maybe correct the sentence - Endornavirus with(?) only one ORF. ...common Endornavirus ...- does not exists, reformulate!

A11: The related contents have been corrected.

Q12: line 160: I do not understand, how resequencing of the 3énd of ORF2 could be proof of ORF2 presence? I expect thorough sequencing of the 3´end of ORF1 for presence/absence of stop codon. Please check and reformulate the sentence. Sanger resequencing of complete genome will be necessary!

A12: Thank you for your suggestion! The presence of ORF2 was confirmed using PCR andSanger sequencing techniques. The sequence between two termination codons were amplified with specific primers based on the 3’ end of ORF1 and ORF2 and sequenced with Sanger method.

Q13: line 190: instead of (Ong et al...) should be [11]. It will be nice to mark Alpha- and Betaendornavirus genera on Figure 3. Furthermore, the term "clade" is not the best one, there are two Genera in the Endornaviridae family (see ICTV report).

A13: Thank you for your suggestion! We have revised the related contents. According to the ICTV reports, Previous study showed that the endornaviruses were divided into clade I (Alphaendornavirus) and clade II (Betaendornavirus) according to the length of the genome, phylogeny of the RdRp, and host type. However, more and more endornaviruses have been found so that the two genera can not be classified well. In the present study, phylogenetic trees indicated that all endornaviruses were clustered into one branch.

Q14: line 193: is it the information, that RsEV1 is the longest endornavirus from R. solani, OR is the longest from all known endornaviruses?

A14: Good suggestion, thnks! RsEV1 is the longest endornavirus from all known endornaviruses. We have revised the sentence.

Q15: lines 262-303. The text is very difficult for people not familiar with the given software and could be simplified. t[1] parameter is not explained.

A15: We have added ‘t[1] represents the first principal component’ to line 259 to explain what is ‘t[1]’.

Q16: line 307:...this is the second (?) ... after CbEVB, CbEC and CbEVG - should be the fourth??

A16: Thank you for your suggestion! We have changed ‘the second’ to ‘the fourth’.

Q17: line 307: should be CbEVC?

A17: Yes! We have revised it.

Q18: lines 312-3: the genus Endornavirus does not exists!

A18: Endornavirus exists according to ICTV reports. So, we keep it.

Q19: lines 312-3: authors declare that RsEV1 containds 2 ORFs, but here wrote: ...classify ... as a member Endornaviridae, because ...containing ONE polyprotein ???

A19: What we describe in this article is that they have many similar genome characteristics, such as containing one large polyprotein encoded by ORF1.

Q20: line 315: superfamily is not defined for Endornaviridae

A20: We have deleted the phrase ‘belonging to the same superfamily’.

Q21: Correct typing errors: line 38-Cryphonectria, 130 - methoxamine..., 136-Agilent.

A21: Thanks! The typing errors have been corrected.

Q22: metabolites in Table 1 - some are capitalized, some not - please unify!

A22: OK, thanks! All the initials of the words in Table 1 have been capitalized.

Q23: lines 381, 395, 447, 449, 454-correct journal abbreviation.

A23: The journal abbreviations have been corrected.

Q24: Line 432 should be ...Bergmüller...,

A24: Bergmuller has been changed to Bergmüller.

Q25: line 433 should be The Golm...

A25: ‘the golm’ have been changed to ‘The Golm’.

Q26: Figure 4- the size of dsRNA does not correspond well with the marker, please use photo with better resolution

A26: Thanks! We have supplemented the experiment and re-uploaded a clear photo.

Q27: Figure 5e - single *, what does it mean?

A27: Sorry for the mistake! Single asterisk (*) indicates P < 0.05. We have added the note to the manuscript.

Reviewer 4 Report

Zheng et al report molecular characterization of a novel endornavirus infecting

Rhizoctonia solani. In addition authors applied GC-MS in metabolomics analysis to test the effect of the virus to its host (instead of RNA-seq, for example). The manuscript is well written and easy to read. 

I warmly recommend this manuscript to be published in viruses.

Author Response

Q: Zheng et al report molecular characterization of a novel endornavirus infecting Rhizoctonia solani. The manuscript is well written and . It was easy to read. I personally liked how GC-MS was applied in metabolomics analysis. I could find only one grammatical oddness. On lines 24-26 I would prefer:

"Pathway annotation revealed that these 32 metabolites were mainly involved in pentose and glucuronate interconversions and glyoxylate, dicarboxylate, starch, and sucrose metabolism."

After grammatical revision I warmly recommend this manuscript to be accepted to be published in viruses.

A: Thanks for your good suggestion! We have revised the grammatical errors on lines 24-26 according to your suggestion.

This manuscript is a resubmission of an earlier submission. The following is a list of the peer review reports and author responses from that submission.

Round 1

Reviewer 1 Report

Zheng et al report molecular characterization of a novel endornavirus infecting Rhizoctonia solani.

The manuscript is well written and . It was easy to read.

I personally liked how GC-MS was applied in metabolomics analysis.

I could find only one grammatical oddness. On lines 24-26 I would prefer:

"Pathway annotation revealed that these 32 metabolites were mainly involved in pentose and glucuronate interconversions and glyoxylate , dicarboxylate,starch, and sucrose methabolism."

After grammatical revision I warmly recommend this manuscript to be accepted to be published in viruses.

Reviewer 2 Report

This MS brings new molecular and biological data about novel endornavirus found in Rhizoctonia solani. Metabolomic profiling of virus-infected and virus-free fungal clones was performed and statistically evaluated.All this data are new a could be interesting for readers. 

Here there are some of my objections and suggestions to improve the text:

I recommend to revise the text by native English speaker, as some not-precise expressions are there and some sentences could be rephrased. 

line 41: Mycoviruses ... are .. RNA genomes ?

line 69: RsEV1- is the first use in the plain text, should be explained

line 87: dsRNAs were ...purified (?) with DNase... - not precise expression

line 97: GC content (of what?) is about...

line 131: ... precipitate with (?) a vortex?  - strange

line 132: MSTFA, TMCS are unknown for biologist, please use full names

line 133: ...centrifuged at 13000 rpm

lines 137-142 describe specific details used in the specific device, could be omitted in the final text

line 154. I wonder that other domains described in another endornaviruses (Helicase, MTR, Cys-rich) were not tested and discussed? see line 315: ...encoding domains...

line 157: UTR is used only one in whole text, should be omitted

line 159: maybe correct the sentence - Endornavirus with(?) only one ORF. ...common Endornavirus ...- does not exists, reformulate!

line 160: I do not understand, how resequencing of the 3énd of ORF2 could be proof of ORF2 presence? I expect thorough sequencing of the 3´end of ORF1 for presence/absence of stop codon. Please check and reformulate the sentence. Sanger resequencing of complete genome will be necessary!

line 190: instead of (Ong et al...) should be [11]. It will be nice to mark Alpha- and Betaendornavirus genera on Figure 3. Furthermore, the term "clade" is not the best one, there are two Genera in the Endornaviridae family (see ICTV report).

line 193: is it the information, that RsEV1 is the longest endornavirus from R. solani, OR is the longest from all known endornaviruses?

lines 262-303. The text is very difficult for people not familiar with the given software and could be simplified. t[1] parameter is not explained.

line 307:...this is the second (?) ... after CbEVB, CbEC and CbEVG - should be the fourth??

line 307: should be CbEVC?

lines 312-3: the genus Endornavirus does not exists!

lines 312-3: authors declare that RsEV1 containds 2 ORFs, but here wrote: ...classify ... as a member Endornaviridae, because ...containing ONE polyprotein ???

line 315: superfamily is not defined for Endornaviridae

Correct typing errors: line 38-Cryphonectria, 130 - methoxamine..., 136-Agilent, 

metabolites in Table 1 - some are capitalized, some not - please unify! 

lines 381, 395, 447, 449, 454-correct journal abbreviation.

Line 432 should be ...Bergmüller..., 

line 433 should be The Golm...

Figure 4- the size of dsRNA does not correspond well with the marker, please use photo with better resolution

Figure 5e - single *, what does it mean? 

Reviewer 3 Report

This manuscript by Zheng et al. is a characterization paper of a new endornavirus, from a strain GD-2 of the rice sheath blast fungus, Rhizoctonia solani AG-1 IA. The complete genome of the virus termed Rhizoctonia solani endornavirus 1 (RsEV1) was determined by a combined approach of NGS and Sanger sequencing technologies. RsEV1 has a few interesting features distinguishing it from many other previously reported endornaviruses. RsEV1 has a large genome (ca. 20 kb) with two ORFs unlike many other endornaviruses with single-ORF genomes, and induces phenotypic alterations such as smaller sclerotia production and hypovirulence in the host plant pathogenic fungus unlike many other endornaviruses showing asymptomatic infections. The symptomatology of RsEV1 was established by comparing isogenic RsEV1-free fungal strain GD-118P and RsEV1-infected fungal strain GD-118P-V1, the second of which was obtained by horizontal transmission. The authors conducted a metabolomics analysis of the two isogenic strains and of 89 metabolites identified, 32 showed differential accumulation, as exemplified by oxalic acid (organic acids), glutamine (amino acids), and maltose (carbohydrates). The authors found a few metabolic pathways, e.g, those for pentose and glucuronate, and starch and sucrose, related to the infection by RsEV1.

As noted by the authors, Ong et al. reported three endornaviruses (Virology, 2016), with a genome organization similar to RsEV1, likely from orchid-asociated fungi, while they were not characterized biologically well. Osaki et al. reported a hypovirulence-conferring endornavirus from another basidiomycete Helicobasidium mompa (Virus Res, 2006). These lower the scientific impact of this study. However, the manuscript is straightforward with new data that provide interesting insights into the endornavirus biology. Another support comes from the fact that papers reporting host metabolome changes caused by fungal viruses on are scarce (Dawe et al., Microbiology, 2009) that should be cited.

This reviewer assessed this paper earlier that was submitted to another virology journal, provided several suggestions for improvement. However, the present manuscript is almost identical to the previous version, and many of my comments have not been incorporated into the current version. Thus, this reviewer has a similar summary of the paper as above, and attached a subset of the same minor suggestions again, hoping that the authors take the suggestions.

Minor points:

Line 69. RsEV1 should be spelled out at its first appearance in the main text.

Line 159. Should be “endornavirus.”

Line 159-160. How did the authors confirm the sequence integrity. By sequencing clones or directly PCR fragments?

Lines 219-231. Did the authors confirm the symptomatology by back-introduction of RsEV1 into GD-118P?

Line 228. Is the increase observed in GD-118P-V1 statistically significant?

Line 307. Why second? Not fourth?

Lines 313 and 330. There are redundant statements.

Line 339. It is not clear to this reviewer how “metabolomics is superior to the traditional biochemistry methods?”

Fig. 3. The tree shows three clades rather than two (page 5). This needs to be discussed

Reviewer 4 Report

In the attached file are reported all my suggestione and comments
